# Nutritional Demand and Consumption Pattern: A Case Study of Pakistan

Naveed Hayat [1], Ghulam Mustafa [1] , Bader Alhafi Alotaibi [2,*] and Abou Traore [3]

[1] Department of Economics, Division of Management and Administrative Science, University of Education, Lahore 54770, Pakistan; naveed.hayat@ue.edu.pk (N.H.); ghulam.mustafa@ue.edu.pk (G.M.)

[2] Department of Agricultural Extension and Rural Society, King Saud University, Riyadh 11451, Saudi Arabia

[3] Department of Community Sustainability, College of Agricultural & Natural Resources, Michigan State University, East Lansing, MI 48824, USA; traoreab@msu.edu

* Correspondence: balhafi@ksu.edu.sa; Tel.: +966-650-424-0201

**Abstract:** Nutrition problems persist over several decades in most developing countries. Poor and vulnerable households in low-income countries do not have sufficient disposable sources to access adequate and diverse diets. This study analyzes the household nutrition consumption patterns in Pakistan at a provincial level. The log-linear Engel's curve approach is applied using household-level data comprising 16,340 households from the Pakistan Household Integrated Expenditure survey (HIES) between 2018 and 2019. The results of the estimated income and household size elasticities reveal that any variation in the households' income brings major changes in their diets, whereas an increase in household size, ceteris paribus (impact of all other factors are held constant), increases the demand for all nutritional intakes except thiamine and cholesterol. Furthermore, the estimated income elasticities for the four provinces provided similarities and differences in the nutritional consumption patterns of households. On the basis of income elasticities, we find the same nutrition pattern in Sindh and Baluchistan and a similar nutrition pattern in Khyber Pakhtunkhwa and Punjab. Finally, the estimated income elasticities show poor nutritional consumption patterns in Sindh and Baluchistan as compared to Punjab and Khyber Pakhtunkhwa. Thus, there is a need to generate nutritional policies in each province, and people should make wise nutrient-dense choices from all food commodities groups. Moreover, this study applies a unique approach to estimate the income elasticities for 17 nutrients using household survey data within the same framework of Engel's curve. The findings of this study have important implications for public policy aiming at malnutrition alleviation and understanding dietary change in Pakistan.

**Keywords:** nutrition consumption; Engel's curve; HIES; elasticities; household

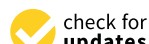



## 1. Introduction

In recent years, studies on the relationship between income and nutrient intake have received significant attention. However, in the available literature, nutrition problems are equated to the inadequacy of energy as measured by the consumption of calories [1]. Thus, the caloric elasticities show only the caloric pattern of the household and could not provide any information about the consumption of important nutrients. A significantly positive association between calories and income does not necessarily mean a higher consumption of nutrients but rather a higher consumption of food items with higher caloric contents. For instance, French et al. [2] are of the opinion that food purchasing patterns may mediate income differences in dietary intake quality. Similarly, when the household income decreases, the household demand for calories changes because the household shifts its preferences to other substitute food items containing a similar number of calories. For example, Colen et al. [3] estimated that less basic and more aspiration diets tend to have higher income elasticities, while elasticities are considerably lower for basic diets. On the other hand, the consumption of necessary nutrients may decrease considerably as

households consume less meat, vegetables, eggs, and milk. There are estimates on the income elasticity for calories, but empirical evidence on the nutrient income elasticity is comparatively limited [4]. Therefore, it is important to explore nutrient income elasticities to guide policy makers in designing appropriate and informed public policies that combat malnutrition and improve diets in these countries [5].

There is an extensive empirical body of literature on household nutrition consumption patterns. For instance, Huang and Lin [6] developed a method for estimating a demand system from household survey data. The results of this study consist of sets of estimated demand elasticities for households. Moreover, the study focused on the treatment of the unit value of nutritional consumption and its relationship with demand elasticity but ignored the treatment of zero-value nutritional consumption. Dhehibi et al. [7] analyzed the Spanish demand for food, considering consumer anxiety about the association between food, diet, and health. However, the study paid no attention to the complementary variables while designing a functional form of the demand system and using a restrictive panel dataset to consider the increasing consumers' anxiety about the relationship between food and health.

Bhattacharya et al. [8] examined the association between nutritional status, poverty, and food insecurity for household members of various age groups. The most outstanding result of the study is that, among school-age children, neither poverty nor food insecurity is associated with nutritional outcomes, while among adults and the elderly, both food insecurity and poverty are predictive. The study explained the association between poverty and low Body Mass Index (BMI) among the elderly and among the young through OLS (Ordinary Least Squares, i.e., the impact of the independent variables on the dependent variable; its purpose is to estimate the unknown parameters in a linear regression model), but this relationship can be explained better by models that consider quality/quantity and time/money trade-offs in food production. Nithya et al. [9] demonstrated the impact of Farming Systems for Nutrition (FSN) interventions on household nutrient intake. Results of the study showed that the intake of nutrients such as protein, vitamin C, iron, and calcium increased significantly in Koraput, and all the nutrients, including energy and vitamin A, increased significantly in Wardha. This evidence showed that the FSN approach improved individual nutrient intake. Emmanuel [4] estimated the income elasticities for various macro and micro-nutritional intakes in rural Mexico. The study also tested the endogeneity between nutritional intakes and income. Results of the study showed that income elasticity for calories is close to zero when controlling for endogeneity. The study tested the endogeneity for calorie intake but could not extend these issues to other nutritional intakes.

Salois et al. [5] estimated income elasticities for calorie and macronutrients such as fat, protein, and carbohydrate intake. Results of the study showed that countries in the upper quantiles have lesser elasticities as compared to the countries in the lower quantiles. The study only incorporated the macronutrients and failed to incorporate the important micronutrients. Uzma [10] focused on the health and nutritional status of children in Pakistani Punjab. The results of the study implied that maternal education and health knowledge are important determinants of child health. The study was limited only to Punjab and could not incorporate the same analysis for other provinces of Pakistan. Salois [11] analyzed the association between income and nutrient intake. Results of the study predict significant growing increases in calorie consumption, which are highly composed of fats. Moreover, this paper focused on macronutrients but did not focus on the deficiency of micronutrients such as iron, vitamin A, iodine, and zinc. Tian and Yu [12] calculated the income elasticities of twenty-two nutrients. This study showed that the elasticities vary across different income groups. The study employed a cross-sectional dataset; the results would have been more substantiated if they had employed a panel dataset. You et al. [13] examined the association between income growth and nutrient intake in urban China. Their study showed that crop income growth raises nutrient intake for malnourished rural households while business and wage income tend to increase urban

households' nutrient intake and reduce nutrition inequality. Instead of a standard panel, the study employed a IV panel model, which required more instruments and variables.

In the above studies, the researchers tried to analyze household nutritional consumption patterns for different countries and regions; some of them also linked household nutrition with other important issues, but none of these studies provided information about the household nutrition consumption pattern at a regional level. Besides, there is limited literature on the demand for nutrients with respect to income in Pakistan.

Pakistan is one of the developing countries where nutrition problems have persisted over several decades. In Pakistan, most poor and vulnerable households do not have sufficient disposable income to access adequate and diverse diets [14]. The culture or lack of knowledge on healthy diets is one of the main causes of nutrition insecurity. Moreover, women and children are more vulnerable to nutritional deficiencies due to their high nutritional requirements as compared to men. In addition, cultural and unbalanced intra-household food practices affect women and children's ability to receive a sufficient share of healthy nutritional intakes in their daily diets [15]. According to Pakistan National Nutritional Survey (NNS) 2018, in the country, about 14.5% of women are underweight, 24% of women are overweight, and 13.9% of women are obese. Moreover, 79.7% of women are Vitamin-D deficient, 27% are Vitamin-A deficient, and 22% of women are zinc deficient, as shown in Table A1 (Appendix A). Similarly, 40.2% of children under 5 years old are stunted, 29% are underweight, and 17.7% are wasted. More than half (53.7%) of Pakistani children are anemic, and 5.7% are severely anemic. The prevalence of anemia is slightly higher (54.2%) amongst boys than girls (53.1%). Children in rural areas are more likely to be anemic (56.5%) than in urban areas (48.9%). Likewise, 51.5% of children are Vitamin-A deficient, 49.1% are iron deficient, 18.6% are zinc deficient, and 62.7% are Vitamin-D deficient, as shown in Table A2 (Appendix A). The statistics of the adolescents (age 10–19 years) showed that in the country, 21% of boys and 12% of girls are underweight, 17.8% of boys and 16.8% of girls are overweight, and 7.6% of boys and 5.5% of girls are obese. Moreover, 54.7% of adolescent girls are anemic, as shown in Table A3 (Appendix A) [16]. The world food crisis of 2008 also shook the nutritional consumption pattern of Pakistani households [17].

In a nutrient-deficient country such as Pakistan, the study of household nutritional consumption patterns has great importance in the form of similarities and dissimilarities in a household's consumption behavior and allocation of income. Moreover, the analysis of household nutritional consumption patterns at provincial levels has become important for the country's nutritional security concerns in the present and future. Therefore, it is needed to check the impact of the income on all food nutrients with different approach, particularly in the context of Pakistan where dietary patterns are changing. It is also very necessary to re-estimate the relationship between income and nutrient intake changes and whether increase in income substantially improves the nutritional status of the people. This may have strong policy implications to alleviate the malnutrition. Furthermore, it can also important to take appropriate steps to counter diet-related non-communicable disease such as obesity that increasing in developing countries like Pakistan. In view of the importance of nutritional consumption patterns and their expected implications for Pakistan, this study provides answers to the question: What are the influencing factors of household nutritional consumption in Pakistan? The main aim of the current study is to analyze the household nutritional consumption pattern in Pakistan at the national and provincial levels. This study has two important contributions; first, it analyzes the nutritional consumption pattern of households at the provincial level; second, it addresses both the micro (vitamins, minerals, etc.) and macro (calories, protein, carbohydrates, fat, fiber, cholesterol, etc.) nutrients, simultaneously.

The study in hands conducted an analysis on disaggregated levels to highlight regional disparities if any. This gives better picture to understand the nutrition consumption patterns and to make policy implications more robust. Investigating ground realities always lessens the burden on scarce resources and gives an edge to informed policy makers to make more effective and suitable policies. Using the household survey data, we estimate the

income elasticities of 17 nutrients, which will lead to more reasonable results for income elasticities of nutrients and will provide better nutritional policy implications in the context of Pakistan at the national and provincial levels. Finally, good nutrition is essential for keeping a healthy life. A healthy diet helps children grow and develop properly and reduces their risk of chronic diseases, including obesity. Adults who eat a healthy diet live longer and have a lower risk of obesity, heart disease, type 2 diabetes, and certain cancers. Healthy eating can help people with chronic diseases manage these conditions and prevent complications. Therefore, the results of this study will provide help to Pakistani households to make wise nutrient-dense choices that enable them to purchase healthier nutritional baskets full of micro and macro nutritional intakes.

## 2. Materials and Methods

### 2.1. Conceptual Framework

At the World Food Summit in 1996, food and nutrition security was defined as the situation "when all individuals or households, at all times, have physical and economic access to sufficient, safe and nutritious food that meets their dietary needs and food preferences for an active and healthy life" [18]. This definition of food and nutrition security reflects two key dimensions: the food and nutrition status and the stability of this food and nutrition status. In our conceptual framework, food availability, food access, and food utilization determine the state of affairs, referred to as the food and nutrition status of an individual or household. Stability refers to two additional important dimensions— vulnerability and resilience towards the state of affairs. It must be stressed that the relation between food and nutrition status and the stability of the food and nutrition status is non-linear and that both categories and their dimensions are highly interlinked [19].

Based on the above discussion, in this study, we test the following hypotheses.

**H1.** *Household income has a positive impact on nutrition demand in Pakistan.*

**H2.** *Household size has a negative impact on nutrition demand in Pakistan.*

### 2.2. Study Site

Pakistan is the world's fifth most populous countries which lies in the South of Asia with a population of about 227 million. The country is at 33rd in term of area with second largest Muslim residents. Its area is spanning 881,913 square kilometers (340,509 square miles). Pakistan is bordered by China to the northeast, Iran to the southwest, Afghanistan to the west, and India to the east. Country's coastal area is 1046 km (650-mile) along the Arabian Sea and Gulf of Oman in the south. Pakistan is the federation of four provinces, namely, Punjab, Sindh, Khyber Pakhtunkhwa (KP), and Baluchistan. Punjab has a population of about 110,000,000, according to the 2017 Pakistan Census. It has more people than the rest of Pakistan combined. According to the Household Integrated Economic Survey (HIES) 2018–2019, the monthly total expenditures/income of households in Punjab is PKR 183,664 (1 USD = PKR 180), and the average household size is approximately six members. Sindh is located in the southeastern region of the country. Sindh is the third-largest province of Pakistan by total area and the second-largest province by population. Sindh has a population of about 47,890,000, according to the 2017 Pakistan Census. According to the HIES 2018–2019, the monthly total expenditures/income of households in Sindh is PKR 182,192 (1 USD = PKR 180), and the average household size is approximately 6.7 members. Khyber Pakhtunkhwa (KP) is situated in the northwestern region of Pakistan, along the Afghanistan–Pakistan border called Durand Line (2640-km) and close to the Tajikistan border. The KP is the third-largest province of the country in terms of both its population and its economy. KP has a population of about 35,530,000, according to the 2017 Pakistan Census. According to the HIES 2018–2019, the monthly total expenditures/income of households in Khyber Pakhtunkhwa is PKR 191,212 (1 USD = PKR 180), and the average household size is approximately 7.3 members. Baluchistan is the largest province in terms of land area, forming the southwestern region of the country, but it is the least

populated province of the country. Baluchistan has a population of about 12,340,000, according to the 2017 Pakistan Census. According to the HIES 2018–2019, the monthly total expenditures/income of households in Baluchistan is PKR 194,058 (1 USD = PKR 180), and the average household size is approximately 7.3 members. To achieve our objective, we selected Pakistan and its four provinces, as shown in Figure 1.

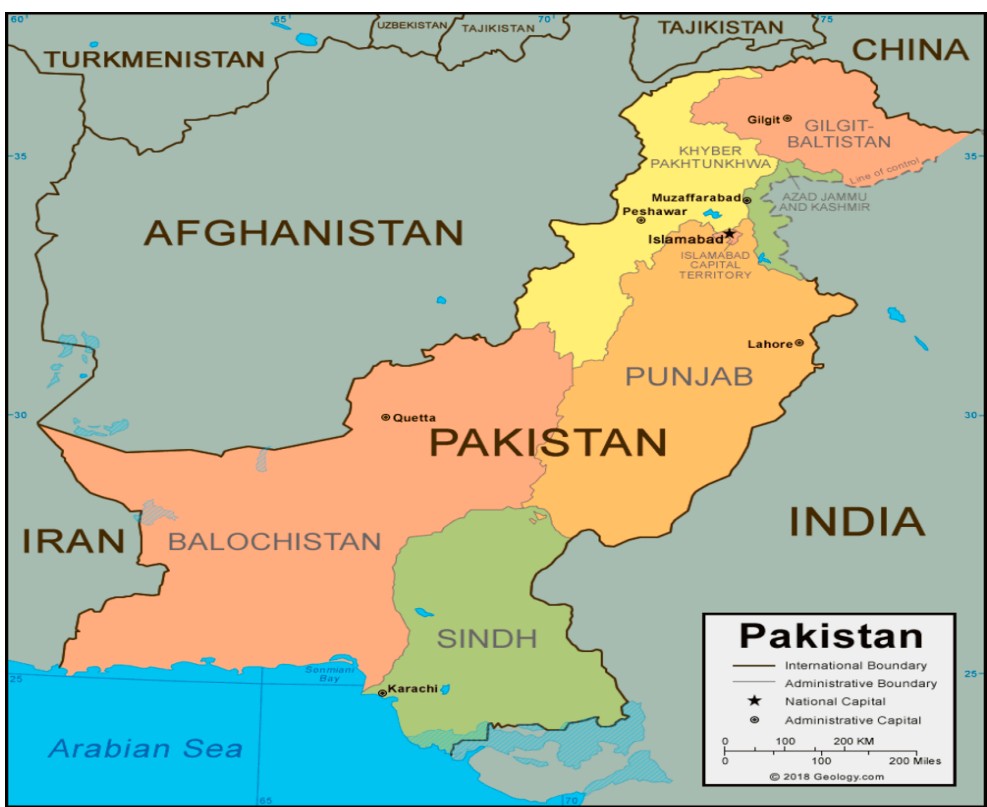

**Figure 1.** Map of Pakistan.

*2.3. Empirical Estimate*

Demand analysis is a science of consumer/household choice/preferences among various goods and services. The analysis of consumer demand is basically the act of analysis of consumer preferences, such as how consumers choose to distribute their income among different goods. The economic theory uses the concept of utility to define the level of satisfaction that comes from the specific distribution of income among various commodities. There are two basic problems of demand analysis: first, how to maximize utility; second, how to minimize expenditures. Let the consumers' utility function depend on food consumption. Therefore, the consumer choice problem can be presented by the following utility maximization problem:

$$MaxU = U(x_1, \ldots, x_n) \tag{1}$$

Subject to:

$$\sum_{i=1}^{n} p_i x_i \leq Y \tag{2}$$

where $U$ is the utility function that depends on $x$ goods $(x_1, \ldots, x_n)$. This utility function is subject to the budget line. Vector $p = (p_1, \ldots p_n)$ and $x = (x_1, \ldots x_n)$ are the prices and quantities of goods and $Y$ represents the consumers' income, respectively. By

solving the first-order conditions and applying the Lagrange multiplier, the following demand function is obtained:

$$x_i^* = x_i(p_1, \ldots\ldots, p_n, Y) \tag{3}$$

The econometric specification of the demand function of Equation (3) provided the Engle curve model [14] such as:

$$x_i^* = a_i + b_i\left(\frac{p_j}{p_i}\right) + c_i\left(\frac{Y}{p_i}\right) + e_i \tag{4}$$

where $x_i^*$ is a demand of the $i$th commodity, $\frac{p_j}{p_i}$ relative price, $\frac{Y}{p_i}$ real income, and $e_i$ random error term. The expenditure on the $i$th commodity is derived by multiplying both sides of Equation (4) by the price of the $i$th commodity:

$$p_i x_i^* = a_i p_i + b_i p_j + c_i Y + e_i p_i \tag{5}$$

where $p_i x_i^*$ is the expenditures of the households on the $i$th commodity. In the empirical literature on household consumption patterns, it has been customary to assume that for every commodity, all households face the same prices. However, this assumption is not true because within a village or urban locality, households can face the same prices, but it can vary among regions. Typically, the household-level cross-sectional dataset did not provide any information about the prices of the commodities. Therefore, it is difficult to capture the impact of price variation on consumption patterns. Therefore, to control the price variation, we assume that all households in the society face the same price structure. Let us assume the same price level for every commodity, then Equation (5) became:

$$E_i = \alpha_i + \beta_i Y + \mu_i \tag{6}$$

where $E_i = p_i x_i$, $\alpha_i = a_i p_i + b_i p_j$, $\beta_i = c_i$, and $\mu_i = e_i p_i$. Equation (6) is the required Engel curve equation, which shows the relationship between expenditure ($E_i$) on commodity $i$ and total income $Y$. Preference ordering may differ from one income group to another or from one household to another. Therefore, we assume an identical utility function for all households or groups of households. Moreover, the Engel curve is a demand function; thus, it must satisfy the general restrictions of demand theory. Since we assume the same price across households, we cannot taste the restrictions of homogeneity and symmetry but can only taste the adding-up restriction. The adding-up restriction implies that:

$$\sum \alpha_i = 0 \; and \; \sum \beta_i = 1$$

Another step of the analysis is to select the appropriate functional form of the Engel curve. The Engel curve has many functional forms, such as linear, double logarithmic, semi-logarithmic, etc.; for our study, we select the general linear and double logarithmic forms of the Engel curve. In the empirical studies of the consumption pattern of various commodities, both measured income and permanent income are used as independent variables [20]. Since permanent income is unobservable and the income data generally suffer from measurement error and may include a transitory component of income, household total expenditures are used as a proxy for the permanent income of the household. Paris [21] emphasized the role of including household size as an explanatory variable in the Engel curve on the grounds that household total expenditure and size are positively correlated, and if the household size in the Engel curve is treated implicitly, then the estimated results may be biased. Moreover, the variations in household size have a comparatively larger effect on the consumption of various commodities than the variation in the total expenditures. The coefficient of the household size captures the effect of economies of scale in consumption among larger households. Houthakker [22] pointed out that the coefficient of the household size represents two effects, namely, a specific effect and an

income effect. The specific effect is the effect resulting from an increase in the demand for different commodities when the household size increases. On the other hand, the income effect refers to the effect when an increase in the household size makes the family relatively poorer in per capita income terms. If the specific effect dominates the income effect, then the coefficient of the household size is positive and vice versa. In this study, we use total household expenditures and household size as explanatory variables. Equation (6) can then be written in the linear and double-logarithmic form as follows:

$$lnE_i = \alpha_i + \beta_i lnY + \beta_j lnHZ + \mu_i \tag{7}$$

Equation (7) is used for estimating the households' consumption pattern. However, in this study, we estimate the household nutrition consumption pattern. Therefore, we modified Equation (7), using the procedure adopted by Tian and Yu [12]. This approach enables us to estimate the household nutrition consumption pattern. Let $N_k$ be defined as the total intake of nutrient $k$ (e.g., calorie, protein, vitamins, etc.) as in [12].

$$N_k = \sum_i C_{ki} * F_i \tag{8}$$

where $C_{ki}$ is the average content of nutrient $k$ in one unit of the $i$th food commodity and $F_i$ is a quantity of the $i$th food commodity consumed. All of these three variables are related to income; thus, we can take the logarithm on both sides of Equation (8) and differentiate it with respect to the logarithm of income $Y$, then the income elasticity of nutrient $k$ can be written as:

$$e_{N_kY} = \sum_i w_{ki} \times e_{F_iY} + \sum_i w_{ki} \times e_{C_{ki}Y} \tag{9}$$

$$w_{ki} = \frac{N_{ki}}{N_k} = \frac{C_{ki} \times F_i}{\sum_i C_{ki} \times F_i} \tag{10}$$

where $e_{N_kY}$ is the elasticity of the $k$th nutrient with respect to income $Y$, is the elasticity of the $i$th food commodity with respect to income $Y$, $e_{C_{ki}Y}$ is the elasticity of average content of nutrient $k$ in one unit of the $i$th food commodity with respect to income $Y$. $N_{ki}$ is the intake of nutrient $k$ from the $i$th food commodity and $w_{ki}$ is the share of the total intake of nutrient $k$ obtained from the $i$th food commodity. As aforementioned, the approach we adopted estimates the nutrient elasticity by converting the food elasticity using a food-to-nutrient conversion factor, which implicitly assumes that the second term in the right side of Equation (9) is zero. The implication of this assumption is that the average contents of every nutrient in each food group are constant. Regarding the high level of aggregation in food demand systems, this is definitely not the case in practice. Since food quality increases with income, it would be true that average nutrient contents, as an index of quality, would certainly change with income. Furthermore, we are more concerned with the relationship between food expenditure elasticity with respect to income and the nutrient elasticity with respect to income. Thus, we continue our deductions as follows:

$$E_i = F_i \times P_i \tag{11}$$

where $E_i$ is the expenditure on the $i$th food commodity and $P_i$ is the average price. From Equation (11), we can easily estimate food expenditure elasticity with respect to income as follows:

$$e_{E_iY} = e_{F_iY} + e_{P_iY} \tag{12}$$

$$e_{F_iY} = e_{E_iY} - e_{P_iY} \tag{13}$$

Substituting Equation (13) into Equation (9) yields:

$$e_{N_kY} = \sum_i w_{ki} \times e_{E_iY} + \sum_i w_{ki} \times e_{C_{ki}Y} - \sum_i w_{ki} \times e_{P_iY} \tag{14}$$

We then define the average cost of nutrient $k$ obtained from the $i$th food commodity as $q_{ki}$, so that the mean price of the $i$th food commodity can be written as:

$$P_i = \frac{q_{ki} \times C_{ki}}{\theta_{ki}} \tag{15}$$

where $\theta_{ki}$ denotes the contribution share of nutrient $k$ to the price of the $i$th food commodity, so that:

$$q_{ki} = \frac{P \times \theta_{ki} C_{ki}}{C_{ki}} \tag{16}$$

Similarly, taking the logarithm of Equation (16) and differentiating it with respect to the logarithm of income, we get:

$$e_{q_{ki}Y} = e_{P_iY} + \left( \frac{dln\theta_{ki}}{dlnC_{ki}} - 1 \right) \times e_{C_{ki}Y} \tag{17}$$

Usually, $\theta_{ki}$, the contribution share of nutrient $k$ to the price of the $i$th food commodity, is taken as an exogenous variable, so that we have $\frac{dln\theta_{ki}}{dlnC_{ki}} = 0$. Then, Equation (17) becomes:

$$e_{q_{ki}Y} = e_{P_iY} - e_{C_{ki}Y} \tag{18}$$

The income elasticity of nutrient $k$ can be written as follows by combining Equations (14) and (18).

$$e_{N_kY} = \sum_i w_{ki} \times e_{E_iY} - \sum_i w_{ki} \times e_{q_{ki}Y} \tag{19}$$

The econometrician effectively assumes that the second term on the right side of Equation (19) is zero; however, this is not the case in practice. The elasticity of the average cost of a nutrient in a food commodity with respect to total expenditure is usually positive. Thus, nutrient elasticity with respect to income, estimated by the approach given in Equation (19), with a high level of aggregation, is overestimated and larger than the direct estimates based on more detailed food data. If there are possible intra-food substitutions in a specific food commodity, the direct nutrient elasticity might be slightly upwardly biased. This bias, however, could be neglected if the number of food commodities is very large. Therefore, we use the direct approach here.

After converting the food consumption into the nutrient intake, a straightforward log-linear Engle curve model is proposed as follows:

$$lnN_k = \alpha_k + \beta_k lnY + \beta_j lnHZ + \mu_k \tag{20}$$

where $Y$ is the household income and $HZ$ is the household size. $\alpha_k$ is the intercept. The $\beta_k$ parameter of the model determines the income elasticity of the $k$th nutrient and whether this nutrient is a luxury, a necessity, or an inferior nutritional intake. For a luxury nutrient $\beta_k > 1$, for a necessary nutrient $\beta_k < 1$, and for an inferior nutrient $\beta_k < 0$. In addition, the $\beta_j$ parameter of the model determines the household size elasticity of the $j$th nutrient. Equation (20) is the Engel curve of nutrients, which will provide income and household size elasticities for each nutrient. Now, with the help of Equation (20), we have easily estimated the nutrition consumption pattern of Pakistani households at the national and provincial levels. The estimation of the Engel curve in Equation (9) for each nutrient is carried out by the seemingly unrelated regression (SUR) method. In SUR, ordinary least square (OLS) is applied to each equation separately. The coefficients obtained in SUR are equivalent to the maximum likelihood coefficients. Furthermore, given the non-substitutable nature of micro-nutrients, the cross-price elasticities should theoretically be equal to zero. Hence, we do not need to include the price information in the econometric model included in Equation (20).

*2.4. Data*

This study used the data of the Pakistan Household Integrated Expenditure Survey (HIES)-2018–2019, conducted by the Pakistan Bureau of Statistics; this dataset comprises 24,809 households. However, for some households, the quantities of various commodities they consumed were missing; therefore, the study used data from 16,340 households. The survey considered includes households' consumption of cereal and cereal products, legumes, vegetables, roots and tubers, spices and condiments, fruits, dairy products, meat and meat products, fish and poultry products, ghee and oil, sugar, sweets, and beverages. These quantities are converted to various nutritional intakes such as calories, protein, fat, carbohydrates, fiber, ash, calcium, phosphorus, iron, zinc, thiamine, riboflavin, niacin, vitamin A, B, and C, and cholesterol, with the help of the food composition table for Pakistan, revised in 2001 [23,24]. Frequency of food items quantities data in HIES 2018–2019 composes three types, i.e., weekly, fortnightly, and monthly. The weekly and 14-day data was first converted into monthly information, and then these groups were joined to construct the household total food amount consumed during the month. This dataset also provides information on the monthly total expenditures made by a household, as well as information on household size. The household total expenditures are the sum of expenses on cereals, grains, other food items, apparel textiles, footwear, fuel, lighting, housing, durable household items, transport, travel, communication, recreation, entertainment, education, medical care, cleaning laundry, and miscellaneous. The household total expenditures are used as a proxy for income because income data generally suffer from measurement errors and may also include a transitory component of income [25].

## 3. Results and Discussion

*3.1. Descriptive Statistics*

Descriptive statistics (Table 1) show household monthly nutrition consumption, monthly total expenditures, and household size. The monthly and daily per capita nutrition consumption is given in Table A4 (Appendix A). The mean monthly household calorie intake in Pakistan is around 362,068 kilocalories. This caloric amount is sensible because the sample we used in this study includes not only adults and adolescents but also children and older people who have lesser energy needs. The mean value of monthly household protein consumption is 6381 g, which is lower than the recommended range. This show that households in Pakistan do not get the required protein from the food they consume. The mean monthly value of fat is 7191 g, which falls in the recommended range. The mean monthly value of carbohydrates is 30,830 g, lower than the recommended range. The amount of fiber consumed by the *i*th household falls within the recommended range. Moreover, the mean value of vitamins shows that Pakistani households consume water-soluble Vitamin C and Vitamin B and fat-soluble Vitamin A in lower than recommended ranges. This indicates that Vitamin A, Vitamin B, and Vitamin C deficiency is still fairly common in Pakistan. As for minerals, the mean value of calcium, phosphorus, and iron are considerably lower than the suggested level, indicating the deficiency of essential minerals in Pakistani household diets. However, the mean value of zinc falls in the recommended range. The coefficients of variation for the mean value of various nutritional intakes showed the largest variation in the consumption of cholesterol and Vitamin A, while the energy consumption exhibited the least variation. In addition, the monthly total expenditures/income of Pakistani households is PKR 186,145.24 (1 USD = PKR 180), and the average household size is approximately seven members. The household composition by age and gender is given in Table A5 (Appendix A).

**Table 1.** Descriptive statistics.

| Variables | Definition | Mean | Standard Deviation | Coefficient of Variation |
|---|---|---|---|---|
| **Dependent** | | | | |
| Nutrient ($N_k$) | Monthly nutrient consumption of the household | | | |
| Energy | Monthly energy consumption of the household (k.cal) | 362,068 | 187,691 | 0.52 |
| Protein | Monthly Protein consumption of the household (g) | 6381 | 4991 | 0.78 |
| Fat | Monthly Fat consumption of the household (g) | 7191 | 18,943 | 2.6 |
| Carbohydrate | Monthly Carbohydrate consumption of the household (g) | 30,830 | 20,530 | 0.66 |
| Fiber | Monthly Fiber consumption of the household (g) | 4032 | 3477 | 0.86 |
| Ash | Monthly Ash consumption of the household (g) | 1709 | 1225 | 0.72 |
| Calcium | Monthly Calcium consumption of the household (mg) | 115,560 | 126,901 | 1.1 |
| Phosphorus | Monthly Phosphorus consumption of the household (mg) | 116,940 | 108,774 | 0.93 |
| Iron | Monthly Iron consumption of the household (mg) | 701 | 422 | 0.60 |
| Zinc | Monthly Zinc consumption of the household (mg) | 1727 | 1161 | 0.67 |
| Thiamine | Monthly Thiamine consumption of the household (mg) | 95 | 112 | 1.2 |
| Riboflavin | Monthly Riboflavin consumption of the household (mg) | 223 | 603 | 2.7 |
| Niacin | Monthly Niacin consumption of the household (mg NE) | 1660 | 1292 | 0.78 |
| Vitamin C | Monthly Vitamin-C consumption of the household (mg) | 2570 | 1838 | 0.72 |
| Vitamin B | Monthly Vitamin-B consumption of the household (mg) | 51 | 74 | 1.5 |
| Vitamin A | Monthly Vitamin-A consumption of the household (mcg RAE) | 44,497 | 178,021 | 4 |
| Cholesterol | Monthly Cholesterol consumption of the household (mg/dL) | 11,691 | 47,002 | 4 |
| **Explanatory** | | | | |
| | Household income and size | | | |
| Total expenditure/income ($Y$) | Monthly expenditure/income of the household (PKR) | 186,145.2 | 136,777.3 | 0.7348 |
| Household size ($HZ$) | Number of family members in the household | 6.7 | 3.1 | 0.4576 |

Source: Computed by authors based on HIES data of Pakistan for the year 2018–2019.

### 3.2. Results of the Estimated Nutrients Engel Curve

The results of the estimated nutrients Engel curve for Pakistan are given in Table 2, while the results of the estimated nutrients Engel curve for the four provinces are given in Table 3 (Punjab), Table 4 (Sindh), Table 5 (Khyber Pakhtunkhwa), and Table 6 (Baluchistan). The results of the diagnostic check (R squared and F statistics) of all the estimated Engel curve regressions are reported in Table A6 (Appendix A). From Table 2 (Pakistan), Table 3 (Punjab), Table 4 (Sindh), Table 5 (Khyber Pakhtunkhwa), and Table 6 (Baluchistan), we found that all estimated income elasticities were positive and highly statistically significant at the 1% level; this implies that the models fit the data very well and all the nutritional intakes are income elastic, such that when household income increases the demand for nutritional intake also increases and vice versa. First, we analyzed the results of Pakistan and then compared the dietary pattern among the four provinces.

For Pakistan (Table 2), the estimated income elasticities ($\beta_k$) of all nutrients except cholesterol were less than one, which implies that these nutrients are necessities, while cholesterol is a luxurious nutritional intake. Cholesterol is more income elastic than other nutritional intakes, indicating that when income rises, demands also rises; it could be justified that due to low purchasing power (poverty) in the country, people are responding more toward the consumption of this nutritional intake as their income changes. These results are in agreement with a previous study in Pakistan [26]. The current results imply that income growth is expected to foster the prevalence of obesity and associated health issues, as indicated by the large income elasticity of cholesterol. Such a trend has been observed in China [27]. This might be one of the reasons to argue that high usage of cholesterol contributes to overweight and obesity problems, which are rapidly increasing in Pakistan [28].

However, the income elasticity of energy is comparatively low, supporting that the energy intake improves slowly with an income growth. This result is acceptable in the Pakistani household, where they experienced a slight but consistent rise in economic growth during the past decade. Thus, the energy elasticity with respect to income falls in the country. Moreover, the income elasticities estimated in this study provide evidence of dietary changes in Pakistani households. The income elasticities of fat and carbohydrates are significantly higher than the income elasticity, meaning that households substitute animal food items, which are rich in protein and fat, with staple food such as cereals enriched with carbohydrates. The significant and positive income elasticities of fiber and water-soluble vitamins such as Vitamin B and C showed a slightly rising consumption of fruits, poultry, and dairy products, which are the main sources of fiber and vitamins. These results can be justified by previous studies [27,29].

**Table 2.** Results of the estimated nutrition Engle curve for Pakistan.

| Nutrient (ln) | Intercept | Income (ln) | Household Size (ln) |
|---|---|---|---|
| Energy | 8.212 *** (0.059) | 0.272 *** (0.005) | 0.678 *** (0.006) |
| Protein | 0.864 *** (0.094) | 0.586 *** (0.008) | 0.366 *** (0.010) |
| Fat | −4.516 *** (0.213) | 0.977 *** (0.019) | 0.218 *** (0.023) |
| Carbohydrate | 2.086 *** (0.083) | 0.623 *** (0.007) | 0.335 *** (0.009) |
| Fiber | 0.521 *** (0.110) | 0.561 *** (0.010) | 0.437 *** (0.012) |
| Ash | −0.383 *** (0.087) | 0.584 *** (0.008) | 0.356 *** (0.010) |
| Calcium | 3.456 *** (0.107) | 0.603 *** (0.010) | 0.376 *** (0.012) |
| Phosphorus | 3.661 *** (0.089) | 0.604 *** (0.008) | 0.307 *** (0.010) |
| Iron | 0.244 ** (0.083) | 0.441 *** (0.008) | 0.480 *** (0.009) |
| Zinc | 1.025 *** (0.071) | 0.455 *** (0.006) | 0.463 *** (0.008) |
| Thiamine | −7.560 *** (0.245) | 0.948 *** (0.022) | −0.015 (0.027) |
| Riboflavin | −1.678 *** (0.103) | 0.504 *** (0.009) | 0.418 *** (0.011) |
| Niacin | 0.278 ** (0.100) | 0.514 *** (0.009) | 0.422 *** (0.011) |
| Vitamin C | 2.843 *** (0.168) | 0.336 *** (0.015) | 0.357 *** (0.018) |
| Vitamin B | −2.940 *** (0.120) | 0.443 *** (0.011) | 0.476 *** (0.013) |
| Vitamin A | −3.204 *** (0.325) | 0.824 *** (0.029) | 0.344 *** (0.035) |
| Cholesterol | −12.952 *** (0.479) | 1.423 *** (0.043) | −0.038 (0.052) |

Source: Computed by authors based on HIES data of Pakistan for the year 2018–2019. *** Significance of the parameters at 1% level of significance, ** significance of the parameters at 5% level of significance. Standard errors of the estimated coefficients are in parentheses.

**Table 3.** Results of the estimated nutrition Engle curve for Punjab.

| Nutrient (ln) | Intercept | Income (ln) | Household Size (ln) |
|---|---|---|---|
| Energy | 8.710 *** | 0.230 *** | 0.679 *** |
|  | (0.077) | (0.007) | (0.009) |
| Protein | 0.648 *** | 0.597 *** | 0.274 *** |
|  | (0.134) | (0.012) | (0.016) |
| Fat | −4.185 *** | 0.938 *** | 0.287 *** |
|  | (0.314) | (0.028) | (0.037) |
| Carbohydrate | 2.327 *** | 0.591 *** | 0.307 *** |
|  | (0.121) | (0.011) | (0.014) |
| Fiber | 0.318 * | 0.565 *** | 0.326 *** |
|  | (0.128) | (0.012) | (0.015) |
| Ash | −0.439 *** | 0.580 *** | 0.282 *** |
|  | (0.123) | (0.011) | (0.015) |
| Calcium | −3.256 *** | 0.606 *** | 0.278 *** |
|  | (0.139) | (0.013) | (0.017) |
| Phosphorus | 3.810 *** | 0.584 *** | 0.296 *** |
|  | (0.133) | (0.012) | (0.016) |
| Iron | 0.189 * | 0.431 *** | 0.430 *** |
|  | (0.114) | (0.010) | (0.014) |
| Zinc | 1.515 *** | 0.404 *** | 0.478 *** |
|  | (0.107) | (0.010) | (0.013) |
| Thiamine | −5.356 *** | 0.764 *** | 0.298 *** |
|  | (0.260) | (0.023) | (0.031) |
| Riboflavin | −1.832 *** | 0.501 *** | 0.338 *** |
|  | (0.139) | (0.012) | (0.017) |
| Niacin | 0.067 | 0.518 *** | 0.316 *** |
|  | (0.137) | (0.012) | (0.016) |
| Vitamin C | 4.460 *** | 0.195 *** | 0.601 *** |
|  | (0.120) | (0.011) | (0.014) |
| Vitamin B | −3.010 *** | 0.470 *** | 0.380 *** |
|  | (0.168) | (0.015) | (0.020) |
| Vitamin A | −1.349 ** | 0.641 *** | 0.614 *** |
|  | (0.458) | (0.041) | (0.055) |
| Cholesterol | −10.394 *** | 1.187 *** | 0.247 ** |
|  | (0.651) | (0.059) | (0.078) |

Source: Computed by authors based on HIES data of Pakistan for the year 2018–2019. *** Significance of the parameters at 1% level of significance, ** significance of the parameters at 5% level of significance, * significance of the parameters at 10% level of significance. Standard errors of the estimated coefficients are in parentheses.

In addition, the positive and comparatively lower-income elasticities of various minerals such as calcium, phosphorus, zinc, and iron show that mineral intake rises gradually with rising income. However, the competitively high-income elasticity of the fat-soluble Vitamin A shows that it does not change much as income grows. It may be due to the fact that Pakistani households already consume too much of it; thus, it is not income sensitive. Observing the household size elasticities ($\beta_j$) for Pakistan (Table 2), we find that, except for thiamine and cholesterol, all household size elasticities of various nutrients were positive and statistically significant. This suggests that when household size increases, the demand for nutrients increases and vice versa. The current results are in line with another study where Novotny et al. [30] found that small household size resulted in lower demands for food.

Comparing the estimated income elasticities of the four provinces, we find the same nutrition patterns in Sindh and Baluchistan and similar nutrition patterns in Khyber Pakhtunkhwa (KP) and Punjab. In KP and Punjab, the estimated income elasticities of all nutrients except cholesterol in Punjab and cholesterol and thiamine in KP were less than one, which implies that these nutrients are necessities while cholesterol is a luxury nutritional intake in Punjab while cholesterol and thiamine are luxury nutritional intakes in KP. Cholesterol and thiamine in these two provinces are more income elastic than

other nutritional intakes. In fact, when the household income rises, the demand for these two nutrients also rises. These results reveal that the households in these two provinces consume diets that are full of nutritional intake. It may be due to the higher purchasing power/income of households, having proper knowledge of selecting nutrient-dense foods in these two provinces.

**Table 4.** Results of the estimated nutrition Engle curve for Sindh.

| Nutrient (ln) | Intercept | Income (ln) | Household Size (ln) |
|---|---|---|---|
| Energy | 8.191 *** (0.142) | 0.265 *** (0.013) | 0.672 *** (0.014) |
| Protein | 1.775 *** (0.143) | 0.528 *** (0.013) | 0.349 *** (0.015) |
| Fat | −5.208 *** (0.423) | 1.041 *** (0.037) | 0.215 *** (0.043) |
| Carbohydrate | 1.923 *** (0.141) | 0.643 *** (0.012) | 0.320 *** (0.014) |
| Fiber | 0.775 *** (0.145) | 0.553 *** (0.012) | 0.329 *** (0.015) |
| Ash | −0.125 (0.134) | 0.580 *** (0.012) | 0.300 *** (0.014) |
| Calcium | 3.645 *** (0.159) | 0.613 *** (0.014) | 0.256 *** (0.016) |
| Phosphorus | 3.823 *** (0.142) | 0.607 *** (0.013) | 0.269 *** (0.014) |
| Iron | 0.960 *** (0.121) | 0.409 *** (0.011) | 0.372 *** (0.012) |
| Zinc | 0.861 *** (0.120) | 0.483 *** (0.011) | 0.369 *** (0.012) |
| Thiamine | −9.289 *** (0.480) | 1.078 *** (0.042) | 0.138 ** (0.049) |
| Riboflavin | −0.922 *** (0.165) | 0.475 *** (0.015) | 0.305 *** (0.017) |
| Niacin | 1.138 *** (0.153) | 0.476 *** (0.014) | 0.331 *** (0.016) |
| Vitamin C | 5.455 *** (0.157) | 0.112 *** (0.014) | 0.630 *** (0.016) |
| Vitamin B | −0.684 *** (0.177) | 0.330 *** (0.016) | 0.408 *** (0.018) |
| Vitamin A | −5.665 *** (0.726) | 1.054 *** (0.064) | 0.288 *** (0.074) |
| Cholesterol | −17.378 *** (1.008) | 1.831 *** (0.089) | −0.123 (0.102) |

Source: Computed by authors based on HIES data of Pakistan for the year 2018–2019. *** Significance of the parameters at 1% level of significance, ** significance of the parameters at 5% level of significance. Standard errors of the estimated coefficients are in parentheses.

Although the estimated income elasticities for Baluchistan are greater than the estimated income elasticities of Sindh, we found similar nutrition patterns in these two provinces. In the province of Sindh and Baluchistan, the estimated income elasticities of all nutrients except fat, thiamine, Vitamin A, and cholesterol were less than one. It implies that these nutrients are necessities while fat, thiamine, Vitamin A, and cholesterol are luxury nutritional intakes. These results indicate that the dietary patterns of households in these two provinces are not according to the recommended standard. Perhaps, the lower purchasing power and the lack of proper knowledge of selecting the nutrient-dense foods of households in these two provinces. Therefore, people of these provinces are encouraged to make wise nutrient-dense choices from all food groups. Studies have found that such choices are the main source of acquiring nutrition, and lacking such attributes makes the calorie and nutrients inelastic [31,32]. Another reason might be the non-availability of food

items full of fat, thiamine, Vitamin A, and cholesterol in these provinces (e.g., Sindh and Baluchistan). The results can be compared with a previous study in Africa that found that income elasticities for vitamins, protein, and fats are higher [33].

The results of the estimated household size elasticities of the four provinces provide interesting insights. For Punjab, the estimated household size elasticities for all nutrients are positive and statistically significant, implying that when household size increases, demand for nutritional intake also increases. It could be justified that when household size increases, nutritional requirements also increase. For Sindh, except for the cholesterol, all other estimated household size elasticities for all nutrients were positively statistically significant. Consequently, when household size increases, nutritional intake demand also increases, while the increase in household size does not affect the demand for cholesterol. A previous study in Pakistan found that small families spend fewer amounts on nutritious food items than big families in rural areas [34]. The results can be justified in the sense that larger families in developing countries have a greater number of working members and, hence, more family income. Therefore, larger-income families consume more nutrients compared to poor families [35,36]. Thus, the study recommends that a program of large income may be an effective policy to improve the nutrition patterns of the country.

**Table 5.** Results of the estimated nutrition Engle curve for KP.

| Nutrient (ln) | Intercept | Income (ln) | Household Size (ln) |
|---|---|---|---|
| Energy | 7.986 *** (0.137) | 0.296 *** (0.019) | 0.688 *** (0.014) |
| Protein | 2.222 *** (0.204) | 0.486 *** (0.019) | 0.417 *** (0.021) |
| Fat | −3.128 *** (0.477) | 0.847 *** (0.043) | 0.336 *** (0.049) |
| Carbohydrate | −0.297 *** (0.180) | 0.556 *** (0.016) | 0.347 *** (0.018) |
| Fiber | 3.845 *** (0.210) | 0.326 *** (0.019) | 0.478 *** (0.022) |
| Ash | 1.543 *** (0.177) | 0.437 *** (0.016) | 0.414 *** (0.018) |
| Calcium | 6.582 *** (0.207) | 0.372 *** (0.019) | 0.434 *** (0.021) |
| Phosphorus | 4.361 *** (0.217) | 0.549 *** (0.020) | 0.351 *** (0.022) |
| Iron | 1.674 *** (0.155) | 0.343 *** (0.014) | 0.500 *** (0.016) |
| Zinc | 1.523 *** (0.164) | 0.424 *** (0.015) | 0.487 *** (0.017) |
| Thiamine | −13.609 *** (0.618) | 1.404 *** (0.056) | −0.026 (0.063) |
| Riboflavin | 0.347 * (0.199) | 0.357 *** (0.018) | 0.448 *** (0.020) |
| Niacin | 2.132 *** (0.182) | 0.377 *** (0.017) | 0.478 *** (0.019) |
| Vitamin C | −5.153 *** (0.451) | 0.968 *** (0.041) | 0.025 (0.046) |
| Vitamin B | −1.440 *** (0.250) | 0.368 *** (0.023) | 0.502 *** (0.026) |
| Vitamin A | −3.111 *** (0.606) | 0.783 *** (0.055) | 0.312 *** (0.062) |
| Cholesterol | −11.217 *** (0.928) | 1.224 *** (0.085) | 0.056 (0.095) |

Source: Computed by authors based on HIES data of Pakistan for the year 2018–2019. *** Significance of the parameters at 1% level of significance, * significance of the parameters at 10% level of significance. Standard errors of the estimated coefficients are in parentheses.

**Table 6.** Results of the estimated nutrition Engle curve for Baluchistan.

| Nutrient (ln) | Intercept | Income (ln) | Household Size (ln) |
|---|---|---|---|
| Energy | 5.563 *** (0.152) | 0.530 *** (0.014) | 0.476 *** (0.015) |
| Protein | −2.215 *** (0.289) | 0.897 *** (0.027) | 0.066 * (0.029) |
| Fat | −9.274 *** (0.750) | 1.453 *** (0.069) | −0.367 *** (0.075) |
| Carbohydrate | −1.286 *** (0.269) | 0.960 *** (0.027) | −0.003 (0.030) |
| Fiber | −3.105 *** (0.443) | 0.951 *** (0.041) | 0.037 (0.045) |
| Ash | −3.568 *** (0.303) | 0.904 *** (0.028) | 0.073 * (0.030) |
| Calcium | 0.206 (0.373) | 0.946 *** (0.034) | 0.035 (0.038) |
| Phosphorus | 0.638 * (0.307) | 0.890 *** (0.028) | 0.070 * (0.031) |
| Iron | −2.839 *** (0.277) | 0.759 *** (0.025) | 0.195 *** (0.028) |
| Zinc | −2.359 *** (0.219) | 0.779 *** (0.020) | 0.214 *** (0.022) |
| Thiamine | −13.143 *** (0.981) | 1.374 *** (0.090) | −0.334 ** (0.099) |
| Riboflavin | −5.073 *** (0.314) | 0.848 *** (0.029) | 0.142 *** (0.032) |
| Niacin | −2.769 *** (0.290) | 0.826 *** (0.027) | 0.147 *** (0.029) |
| Vitamin C | −5.050 *** (0.842) | 0.972 *** (0.077) | 0.199 * (0.085) |
| Vitamin B | −4.946 *** (0.373) | 0.700 *** (0.034) | 0.213 *** (0.037) |
| Vitamin A | −12.207 *** (1.230) | 1.660 *** (0.113) | −0.310 * (0.124) |
| Cholesterol | −26.680 *** (2.103) | 2.593 *** (0.193) | −0.567 ** (0.212) |

Source: Computed by authors based on HIES data of Pakistan for the year 2018–2019. *** Significance of the parameters at 1% level of significance, ** significance of the parameters at 5% level of significance, * significance of the parameters at 10% level of significance. Standard errors of the estimated coefficients are in parentheses.

For KP (Table 5), except for thiamine, Vitamin C, and cholesterol, all other estimated household size elasticities and nutrients were positive and statistically significant. When household size increases, the demand for nutritional intake also increases, while the increase in household size does not affect the demand for thiamine, Vitamin C, and cholesterol.

For Baluchistan, the estimated household size elasticities for energy, protein, ash, phosphorus, iron, zinc, riboflavin, niacin, Vitamin B, and Vitamin C were positive and statistically significant, indicating that when household size increases, the demand for these nutritional intakes also increases (Table 6). The previous study also found a positive and significant influence of household size on the amount of food consumed. For instance, Chiaka et al. [37] estimated that there is a significant relationship of 0.636 between household size and food expenditure using the Pearson correlation coefficient. So, the results are in agreement with a previous study. The estimated household size elasticities of fat, thiamine, Vitamin A, and cholesterol were negative and significant. There was a strong indication that the increase in household size decreases the demand for these nutritional intakes. Moreover, the estimated household size elasticities of carbohydrates, fiber, and calcium were not statistically significant; thus, the increase in household size does not affect the demand for these nutritional intakes.

In a nutshell, the nutritional consumption pattern of Pakistan concluded through the current empirical analyses suggests that changes in the household income cause major changes in their diets. On the other hand, a large increase in the demand for cholesterol can be expected following an increase in household income. An increase in household size, ceteris paribus, increases the demand for all nutritional intakes except for thiamine and cholesterol. On the basis of income elasticities, we find exactly the same nutrition pattern in Sindh and Baluchistan and a similar nutrition pattern in KP and Punjab. Cholesterol in Punjab and cholesterol and thiamine in KP are more income elastic than other nutritional intakes, while in Sindh and Baluchistan, fat, thiamine, Vitamin A, and cholesterol are more income elastic than other nutritional intakes. This shows that the dietary pattern of households in Punjab and KP is better than the dietary pattern of households in Sindh and Baluchistan. Moreover, the increase in the household size in Punjab increases the demand for all nutritional intakes. The increase in the household size in KP also increases the demand for all nutritional intakes except thiamine, Vitamin C, and cholesterol. On the other hand, the increase in the household size in Sindh increases the demand for all nutritional intakes except cholesterol. The increase in the household size in Baluchistan elevates the demand for energy, protein, ash, phosphorus, iron, zinc, riboflavin, niacin, Vitamin B, and Vitamin C, while the demand for fat, thiamine, Vitamin A, and cholesterol decreases. Furthermore, there is a high level of inequality in protein and vitamins among the people of Pakistan and within the provinces. For instance, high usage of protein and Vitamin C caused obesity among the households and its compositions. A healthy diet is recommended, which is not affordable for the majority of people [38]. Hence, income should be increased for balanced dietary intake.

## 4. Conclusions

This study analyzes the nutritional consumption pattern of Pakistani households at a provincial level. The study is built upon HIES for the year 2018–2019, and for the empirical estimation, the study applied the log-linear Linear Engel's curve approach. The nutritional consumption pattern of Pakistan concluded through the current empirical analyses suggests that changes in the household income cause major changes in their diets. An increase in household size, ceteris paribus, increases the demand for all nutritional intakes except for thiamine and cholesterol. On the basis of income elasticities, we find exactly the same nutrition pattern in Sindh and Baluchistan and a similar nutrition pattern in KP and Punjab. Furthermore, we find that the dietary pattern of households in Punjab and KP is better than the dietary pattern of households in Sindh and Baluchistan. Therefore, the study recommends that a large income may be a better option to improve the nutritional status of the country as compared to other product-based subsidy policies. An increase in household income through poverty alleviation programs that provide direct cash transfer, such as the recently introduced Ehsaas (subsidy) Program of the Government of Pakistan, would be successful in achieving an increase in consumption of the key nutrients at the household level in Pakistan. Moreover, descriptive statistics of the study reveal that Pakistan is a food deficit country with respect to most nutrients. Hence, the development of a strong connection between the food and nutritional insecurity in the ministries of the four provinces with the federal ministry of food and nutritional insecurity of Pakistan will also be helpful. The overall collaboration will improve the nutritional consumption pattern of households. According to our result, household size is one of the most important determinants of the nutritional demand; therefore, diverse population control measures may improve the nutritional standard of Pakistani households. The current study estimated the nutritional demand at a provisional level; however, future research can further narrow and can incorporate consumption patterns at a district level. Moreover, nutritional demand projections can also be completed in future research for better policy formulation.

**Author Contributions:** Conceptualization, N.H., G.M. and B.A.A.; methodology, N.H. and G.M.; software, N.H., G.M. and A.T.; validation, N.H.; formal analysis, G.M.; investigation, B.A.A. and G.M.; resources, B.A.A.; data curation, N.H. and G.M.; writing—original draft preparation, G.M.;

writing—review and editing, N.H. and A.T.; visualization, B.A.A.; supervision, N.H. and B.A.A.; project administration, N.H.; funding acquisition, A.T. All authors have read and agreed to the published version of the manuscript.

**Funding:** This research was funded by Researchers Supporting Project Number (RSP2022R443), King Saud University, Riyadh, Saudi Arabia.

**Institutional Review Board Statement:** Not applicable.

**Informed Consent Statement:** Not applicable.

**Data Availability Statement:** Data supporting reported results will be made available to the interested researchers upon request.

**Acknowledgments:** The authors are grateful to the Deanship of Scientific Research and RSSU at King Saud University for their technical support.

**Conflicts of Interest:** The authors declare no conflict of interest.

## Appendix A

**Table A1.** Nutritional status and micronutrient deficiencies in women of reproductive age 15–49 years in Pakistan.

| Nutrition status | | | |
|---|---|---|---|
| Underweight prevalence | | 14.5% | |
| Overweight prevalence | | 24.2% | |
| Obesity prevalence | | 13.9% | |
| Micronutrient deficiencies | Non-pregnant | Pregnant | Overall |
| Vitamin D deficiency | 79.6% | 81.2% | 79.7% |
| Vitamin A deficiency | 30% | 27% | 27% |
| Zinc deficiency | 21.1% | 37.5% | 22.1% |
| Iron deficiency | 33.6% | 46.9% | 34.3% |
| Anemia | 43% | 35.5% | 42.6% |
| Calcium deficiency | 16.2% | 32.6% | 26.5% |
| Folic Acid deficiency | 45.3% | 44.5% | 44.5% |
| Vitamin B12 deficiency | 19.5% | 32.3% | 20.3% |

Source: Government of Pakistan Nutritional Survey 2018–2019.

**Table A2.** Nutritional status and micronutrient deficiencies in children age 6–59 months in Pakistan.

| Nutrition status | Boys | Girls | Overall |
|---|---|---|---|
| Underweight prevalence | 29.3% | 28.4% | 28.9% |
| Stunting prevalence | 40.9% | 39.4% | 40.2% |
| Wasting prevalence | 18.4% | 17% | 17.7% |
| Overweight prevalence | 9.7% | 9.2% | 9.5% |
| Micronutrient deficiencies | | | |
| Vitamin D deficiency | 62.3% | 63% | 62.7% |

**Table A2.** *Cont.*

| Nutrition status | Boys | Girls | Overall |
|---|---|---|---|
| Vitamin A deficiency | 51.6% | 51.3% | 51.5% |
| Zinc deficiency | 18.8% | 18.4% | 18.6% |
| Iron deficiency | 50% | 48.2% | 49.1% |
| Anemia | 54.2% | 53.1% | 53.7% |
| Calcium deficiency | 32% | 32.4% | 32.2% |
| Folic Acid deficiency | 34.3% | 35.5% | 34.9% |
| Vitamin B12 deficiency | 26% | 24.1% | 25.1% |

Source: Government of Pakistan Nutritional Survey 2018–2019.

**Table A3.** Nutritional status and micronutrient deficiencies in adolescents age 10–19 years in Pakistan.

| Nutrition status | Boys | Girls |
|---|---|---|
| Underweight prevalence | 21.1% | 11.8% |
| Short stature prevalence | 31.7% | 28.5% |
| Overweight prevalence | 17.8% | 16.8% |
| Obesity prevalence | 7.6% | 5.5% |
| Micronutrient deficiencies | | |
| Anemia | - | 54.7% |

Source: Government of Pakistan Nutritional Survey 2018–2019.

**Table A4.** Per capita monthly and daily consumption of nutrients.

| Variables | Monthly Per Capita Consumption | Daily Per Capita Consumption |
|---|---|---|
| Nutrient ($N_k$) | Mean | Mean |
| Energy (k.cal) | 56,721 | 1891 |
| Protein (g) | 1034 | 34 |
| Fat (g) | 1163 | 39 |
| Carbohydrate (g) | 5013 | 167 |
| Fiber (g) | 640 | 21 |
| Ash (g) | 277 | 9 |
| Calcium (mg) | 18,666 | 622 |
| Phosphorus (mg) | 19,190 | 640 |
| Iron (mg) | 113 | 3.8 |
| Zinc (mg) | 286 | 9.5 |
| Thiamine (mg) | 17 | 0.56 |
| Riboflavin (mg) | 45 | 1.5 |
| Niacin (mg NE) | 276 | 9.2 |
| Vitamin C (mg) | 425 | 14 |
| Vitamin B (mg) | 9 | 0.31 |
| Vitamin A (mcg RAE) | 7130 | 238 |
| Cholesterol (mg/dL) | 1877 | 63 |

Source: Computed by authors based on HIES data of Pakistan for the year 2018–2019.

**Table A5.** Household composition by age and gender.

| Composition | Definition | Mean | Standard Deviation | Coefficient of Variation |
|---|---|---|---|---|
| Children (age ≤ 9) | Number of children in the household | 1.78 | 1.71 | 0.96 |
| Adolescent (age 10–19) | Number of adolescents in the household | 1.65 | 1.60 | 0.97 |
| Adult (age > 19) | Number of adults in the household | 3.23 | 1.71 | 0.53 |
| Male | Number of male members in the household | 2.48 | 1.51 | 0.61 |
| Female | Number of female members in the household | 2.39 | 1.40 | 0.58 |

Source: Computed by authors based on HIES data of Pakistan for the year 2018–2019.

**Table A6.** Diagnostic check (R squared and F statistics) of the estimated Engel curve regressions.

| | Pakistan | | Punjab | | Sindh | | Khyber Pakhtunkhwa | | Baluchistan | |
|---|---|---|---|---|---|---|---|---|---|---|
| Nutrient | R-Squared | F-Statistics | R-Squared | F-Statistics | R-Squared | F-Statistics | R-Squared | F-Statistics | R-Squared | F-Statistics |
| Energy | 0.60 | 12,283 *** | 0.61 | 5443 *** | 0.49 | 1993 *** | 0.68 | 3126 *** | 0.73 | 3194 *** |
| Protein | 0.40 | 5510 *** | 0.39 | 2186 *** | 0.47 | 1830 *** | 0.45 | 1212 *** | 0.47 | 1015 *** |
| Fat | 0.20 | 2049 *** | 0.20 | 869 *** | 0.21 | 538 *** | 0.22 | 421 *** | 0.19 | 279 *** |
| Carbohydrate | 0.47 | 7290 *** | 0.45 | 2791 *** | 0.54 | 2399 *** | 0.52 | 1622 *** | 0.47 | 1016 *** |
| Fiber | 0.34 | 4230 *** | 0.41 | 2369 *** | 0.47 | 1847 *** | 0.38 | 885 *** | 0.29 | 467 *** |
| Ash | 0.43 | 6226 *** | 0.42 | 2523 *** | 0.52 | 2204 *** | 0.49 | 1425 *** | 0.45 | 947 *** |
| Calcium | 0.35 | 4457 *** | 0.38 | 2098 *** | 0.44 | 1602 *** | 0.38 | 918 *** | 0.36 | 664 *** |
| Phosphorus | 0.42 | 5843 *** | 0.39 | 2207 *** | 0.50 | 2019 *** | 0.43 | 1106 *** | 0.43 | 894 *** |
| Iron | 0.42 | 5993 *** | 0.42 | 2466 *** | 0.49 | 1945 *** | 0.55 | 1786 *** | 0.46 | 976 *** |
| Zinc | 0.50 | 8304 *** | 0.45 | 2877 *** | 0.54 | 2406 *** | 0.56 | 1875 *** | 0.59 | 1693 *** |
| Thiamine | 0.13 | 1183 *** | 0.21 | 904 *** | 0.17 | 419 *** | 0.23 | 446 *** | 0.11 | 147 *** |
| Riboflavin | 0.34 | 4121 *** | 0.33 | 1736 *** | 0.35 | 1090 *** | 0.40 | 987 *** | 0.42 | 856 *** |
| Niacin | 0.35 | 4437 *** | 0.34 | 1808 *** | 0.40 | 1340 *** | 0.48 | 1333 *** | 0.45 | 967 *** |
| Vitamin C | 0.09 | 834 *** | 0.34 | 1748 *** | 0.35 | 1114 *** | 0.22 | 416 *** | 0.12 | 164 *** |
| Vitamin B | 0.26 | 2880 *** | 0.25 | 1167 *** | 0.28 | 790 *** | 0.33 | 733 *** | 0.30 | 488 *** |
| Vitamin A | 0.08 | 745 *** | 0.09 | 328 *** | 0.09 | 198 *** | 0.13 | 224 *** | 0.11 | 144 *** |
| Cholesterol | 0.08 | 693 *** | 0.08 | 299 *** | 0.11 | 239 *** | 0.10 | 160 *** | 0.09 | 117 *** |

Source: Computed by authors based on HIES data of Pakistan for the year 2018–2019. *** Significance of the parameters at 1% level of significance.

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
