# Peer review of "Nutritional Demand and Consumption Pattern: A Case Study of Pakistan"

_sustainability, doi:10.3390/su14127068_

Round 1

Reviewer 1 Report

the authors properly answered to the reviewer's comments. 

Author Response

Comments and Suggestions for Authors:

Comment:

The authors properly answered to the reviewer's comments.

Response:

Thank you very much for accepting the changes. You suggestions really improved the quality of paper

Reviewer #1:

Yes

Can be improved

Response

Is the content succinctly described and contextualized with respect to previous and present theoretical background and empirical research (if applicable) on the topic?

(x)

( )

Thank you very much

Are the research design, questions, hypotheses and methods clearly stated?

( )

(x)

It is further improved

Are the arguments and discussion of findings coherent, balanced and compelling?

(x)

( )

Thank you very much

For empirical research, are the results clearly presented?

( )

(x)

It is further elaborated. New studies are added. Line 569-573 and 602-607

Is the article adequately referenced?

(x)

( )

Thank you very much

Are the conclusions thoroughly supported by the results presented in the article or referenced in secondary literature?

(x)

( )

Thank you very much

Reviewer 2 Report

The authors present a paper on an interesting topic, to which I add a few remarks:
- On line 242, the authors use the same symbol for quantity and demand function (x).
- In Chapter 3.1, the authors could provide more detailed characteristics of the household, particularly its composition (they only report the average number of household members).
- My most important remark concerns Table 1. The values given seem incorrect. For example, the monthly protein consumption in a household is 212 kg, which means that with an average household size of 6.7 members, the consumption is 1.05 kg of pure protein per person. Alternatively, for vitamin C, the monthly consumption of 0.8566 kg is based on daily consumption of 4.2 grams (the recommended daily dose is 80-100 milligrams per day). These results undermine confidence in the accuracy of other results.

Author Response

We incorporated all changes. thank you very much for reviewing the paper.

Reviewer 3 Report

Dear Authors,

The subject of your article (nutritional demand and consumption pattern) is actual and, will remain actual many years from now in some countries where access to food resources is scarce through many factors.

The research released in the article could be useful for the decision of the government people to vision a better law release to help people in need.

The article has an interesting view of data, but needs a well-illustration for a better and easy understanding of the presented research, even for the random reader.

 Only a few points should be addressed by the authors:

  • Line 5 to 12 - For author 3, please complete the mail address and also, for all authors, specified the initial of the names (N.G. and so on).
  • Line 75 – please explain OLS
  • Table 1. – I consider that are sufficient two numbers after the point, for all presented data (mean - 212.7016 as 212.70, standard deviation - 166.35475 as 166.35 and so on, to be easier to follow)
  • Table 1. – please explain the high value of standard deviation - 631.44754, compared with mean - 239.7159 – for variable fat
  • Please explain – the number of family members in the household – a typical family in Pakistan is formed of parents and children or grandparents, parents and children
  • Please explain - ceteri paribus – found in the abstract section, Line 554, and in the Conclusions section
  • Figure 2 – Map of Pakistan – it could be very helpful the illustrate for each region the population, the income, the number of the family member, and other important factors that make more easy to understand the Tables 2 to 6.

Thank you!

Author Response

We incorporated all changes. Thank you very much for reviewing the paper. Please see the attachment.

Round 2

Reviewer 2 Report

The authors did not deal at all with the nonsense of the presented values and were satisfied with the statement that the people of Pakistan are overweight.

If a household consumed 212.7 kg of protein, 239.72 kg of fat and 1027.65 kg of carbohydrates per month, this would correspond to an energy intake of 212.7 kg * 4000 kcal/kg + 239.72 kg * 9000 kcal/kg + 1027,65 kg * 4000 kcal/kg = 7118880 kcal. In Table 1, the authors state the monthly household consumption of 362068 kcal (which is a realistic value of 1809 kcal per person per day).

Author Response

Dear reviewer, special thanks for this comment, we have revised the descriptive statistics (Table 1). The value of each nutrient consumed by the households is now given in its corresponding unit (i.e. K.cal, g, mcg, mg NE etc.). We also reported the per capita monthly and daily consumption of nutrients in Table D4 (appendix). After changing the nutrients values into their original units, we again estimate our regressions and revised the regression results (see Table2, Table 3, Table 4, Table 5, and Table 6). Your comment really helped us a lot. Please suggest us if further improvements are required.      

Round 3

Reviewer 2 Report

No more comments.

This manuscript is a resubmission of an earlier submission. The following is a list of the peer review reports and author responses from that submission.

Round 1

Reviewer 1 Report

This manuscript deals with the study of the nutritional consumption pattern of Pakistani households at 394 a provincial level. The work is interesting but needs a strong revision.

Comments

  • Please revise the English in the whole paper. Some typo corrections have been made by the reviewer, but the others have to be made by the authors.
  • Page 1, Line 34: Please avoid the repetition of the word “nutrition”
  • Page 1, Line 39: “…caloric contents”. Please provide a reference for this sentence with some examples.
  • Page 1, Line 42-44: what the authors write may be intuitive and logical. however, for proper scientific work, the claims made must be correlated with numbers. therefore, the authors should report some supporting references
  • Page 1, Line 44-45: please avoid the repetition of the word “Thus”.
  • Page 2, Line 49-50: please provide a reference
  • Page 2, Line 50-56: like the comment above, the claims made must be correlated with numbers or proper surveys. therefore, the authors should report some supporting references
  • Page 2, Line 56-70: the authors provide a lot of data and percentages. The reviewer suggests re-organizing the numbers in a table in order to be easy to read and understand.
  • Page 2, Line 73: please avoid the repetition of the word “nutrition”
  • The reviewer suggests to incorporate Section 2 “literature review” with the introduction. Moreover, the reviewer suggests to stress more the novelty related to the analysis of mineral compounds needed by people and not only by the novelty of the analysis at a regional level (it is not so original in the opinion of the reviewer).
  • Please correct the position of the number of the equations in the whole paper.
  • Page 4, Line 150: explain better what the utility U is and the “budget line”
  • Page 4, Line 160: “The expenditure on commodity i is derived as follows”: how can you derive i from equation 5? It is not a variable but just a subscript…please explain better.
  • Page 4: what are alfa, beta, a, b, c etc à please report the values of these parameters
  • Page 4, Line 164: what are the disadvantages of assuming the same price level for every commodity?
  • Please read again and change the whole paragraph “3.1 Empirical Estimate” because it is not clear. The suggestion is to explain each choice of the equation, to report what is every parameter (e.g. creating a “Nomenclature” section) making in this way the paragraph clearer.
  • Page 5, line 228: “Fibber”à “Fiber” or “Fibre”
  • Page 5, line 235: “are the sum of expenditures on”à you can remove the word “expenditures” in order to avoid a repetition.
  • Page 5, line 249: “consumed by t households” à what is t?
  • Please revise the format of table 1

Please revise the conclusion. Are too long and there are some comments that can be moved from “conclusion” to “results and discussion” section.

Reviewer 2 Report

It would be appropriate to give it under Table 1 as an international currency.
It is important to present the consumption per capita.

Reviewer 3 Report

General comments

This study focuses on the household nutrition consumption patterns in Pakistan at a provincial level. The topic is very interesting and it has practical importance, particularly for Pakistan and other developing countries. Though this topic is quite interesting and of practical significance, the manuscript suffers from poor presentation style, introduction, weak results, and discussion section. The author should work on it to improve its quality.

Title: The title should be “Nutritional Demand and Consumption Pattern: A case study of Pakistan” for meaningful presentation.

Abstract

It is not well structured. The author should add a little bit more about the measurement approaches of nutritional demand and consumption pattern, sample size, key finding, and practical implications.

Introduction

-The introduction is poor and not structured. The author failed to introduce the issue, its importance, and its practical implication.

-The author could not focus on the research gap and specific objective.

-There is no coherence in the description.

-Need to avoid unrelated discussions.

-What is the research question of this study?

-How will this study be beneficial for society?

-The author should at least two paragraphs to address the above issues. Otherwise, the presentation will fail to convey the study's key message.

-There is a huge inconsistency in the text under the introduction section.

Literature review

This section fails to provide the exact situation and logic of the study. It is also poor. The author is advised to include:

Why are the nutritional demand and consumption patterns serious issues to study? -How do you draw the variables from the study?

-How do you draw the hypothesis of the study?

-Where is the research gap?

Materials and Methods

-The methodology section is not complete.

It is not understandable why the author starts this section by explaining the econometric model?

-There are only 2 sub-sections that fail to explain the method properly. The author should include few points:

-What is the nutritional status and consumption pattern existing in the study areas?

-What are the geographical features of the study areas?

-How do you select the variables?

- How do you the indicator and scale for measuring variables?

-What statistical approach you use and why? Actually, there is no description of a statistical test.

-The overall section is very poor, and not reproducible. So, this section is not acceptable.

Results

-How can we find out results without proper methodology?

-This section is also very poor and haphazard.

-There is no planning for presenting the findings properly.

-The description lacks coherence due to procedural weakness.

-The author is advised to revise the whole section by following new methodology.

Discussion

-How can we find out discussions without proper methodology?

-This section is also very poor and haphazard.

-There is no planning for presenting the findings properly.

-There is a little connection between research objective and findings.

-The author is advised to revise the whole section by following new methodology.

Conclusion

This section is not structured. It should be revised by adding key findings, recommendations and practical implications.

References

Need to check the whole section and follow the journal style.